# Population Pharmacokinetic Analysis of Pazopanib in Patients and Determination of Target AUC

**DOI:** 10.3390/ph14090927

**Published:** 2021-09-15

**Authors:** Agustos Cetin Ozbey, David Combarel, Vianney Poinsignon, Christine Lovera, Esma Saada, Olivier Mir, Angelo Paci

**Affiliations:** 1Service de Pharmacologie, Département de Biologie et Pathologie Médicales, Gustave Roussy, Université Paris-Saclay, F-94800 Villejuif, France; ozbey.acetin@gmail.com (A.C.O.); david.combarel@gustaveroussy.fr (D.C.); vianney.poinsignon@gustaveroussy.fr (V.P.); 2Service de Pharmacocinétique, Faculté de Pharmacie, Université Paris-Saclay, F-92296 Chatenay-Malabry, France; 3Centre Antoine Lacassagne, Délégation à la Recherche Clinique, F-06189 Nice, France; christine.lovera@nice.unicancer.fr (C.L.); Esma.SAADA-BOUZID@nice.unicancer.fr (E.S.); 4Département de Soins Ambulatoire, Gustave Roussy, F-94800 Villejuif, France; olivier.mir@gustaveroussy.fr

**Keywords:** cancer, tyrosine kinase inhibitors, population pharmacokinetics, therapeutic drug monitoring (TDM)

## Abstract

Pazopanib is a potent multi-targeted kinase inhibitor approved for the treatment of advanced renal cell carcinoma and soft tissue sarcoma. The pharmacokinetics of pazopanib is characterized by a significant inter- and intra-patient variability and a target through plasma concentration of 20.5 mg·L^−1^. However, routine monitoring of trough plasma concentrations at fixed hours is difficult in daily practice. Herein, we aimed to characterize the pharmacokinetic (PK) profile of pazopanib and to identify a target area under the curve (AUC) more easily extrapolated from blood samples obtained at various timings after drug intake. A population pharmacokinetic (popPK) model was constructed to analyze pazopanib PK and to estimate the pazopanib clearance of a patient regardless of the time of sampling. Data from the therapeutic drug monitoring (TDM) of patients with cancer at Institute Gustave Roussy and a clinical study (phase I/II) that evaluates the tolerance to pazopanib were used. From the individual clearance, it is then possible to obtain the patient’s AUC. A target AUC for maximum efficacy and minimum side effects of 750 mg·h·L^−1^ was determined. The comparison of the estimated AUC with the target AUC would enable us to determine whether plasma exposure is adequate or whether it would be necessary to propose therapeutic adjustments.

## 1. Introduction

Currently, there are 88 small molecule kinase inhibitors (SMKIs) approved by the FDA and by other regulatory agencies. Their main application area is oncology; however, one-third of the SMKIs in clinical development address different kinds of disorders such as rheumatoid arthritis. The latest clinical trial of SMKIs shows that the approximately 45 targets of approved kinase inhibitors represent only about 30% of the human kinome, which indicates that there are still substantial unexplored opportunities for this drug class [1].

Pazopanib (Figure 1), a multi-kinase inhibitor (MKI), was approved by the FDA (Food and Drug Administration) and by the EMA (European Medicines Agency) for the treatment of advanced renal cancer or soft tissue sarcoma in 2009 and 2013, respectively. Pazopanib targets VEGFR1, VEGFR2, VEGFR3, PDGFR, FGFR, and c-Kit with IC_50_ of 10, 30, 47, 84, 74, and 140 nM, respectively [2].

Pazopanib is a film-coated, immediate release (IR) tablet available at two dosage forms, 200 and 400 mg. The recommended dosage is 800 mg QD per os [3]. After an oral administration of single 800 mg dose (n = 10) the plasmatic concentration (C_max)_ [CV%] was 19.46 [176%] mg·L^−1^. At steady state after 22 day (n = 10), the C_max_ was 45.1 [68.8%] mg·L^−1^ [2,4]. Furthermore, pazopanib exhibits poor bioavailability, ranging from 14% to 39% and, after administrating 800 mg, the area under curve (AUC_0__→__∞_) is 650 ± 500 µg·h·mL^−1^ [3,5].

Pazopanib exhibits extensive binding to plasma proteins (>99%) and is a substrate for P-glycoprotein (P-gp, MDR1, ABCB1) and breast cancer resistance protein (BCRP, ABCG2). Pazopanib is mainly metabolized by cytochrome P450 CYP3A4, and to a lesser extent, by CYP1A2 and CYP2C8. One of the metabolites (metabolite M26 or GSK1268997) (Figure 2), inhibits the proliferation of human umbilical vein endothelial cells stimulated by VEGF with a potency similar to that of pazopanib but accounts for less than 10% of the total drug. The other metabolites are 10–20 times less active. Therefore, the activity of pazopanib is mainly dependent on the parent pazopanib molecule [3].

After once-daily administration, pazopanib has a half-life of 30.9 h and reaches a steady state in the body within 7 to 8 days. It is mainly eliminated in the feces (65% of parent drug), urinary elimination being less than 4%. Previous studies have shown a correlation between a trough plasma concentration of 20 mg·L^−1^ and an increase in progression-free survival (PFS), making individualized pazopanib dosing a promising approach to improve outcome in terms of safety and efficacy [6,7].

One of the most important challenges in treatment with MKIs, and by extension, in pazopanib treatments, is the management of inter and intra-patient variability. Indeed, several studies have shown a high variability of the concentrations of pazopanib in the blood, which may be affected by various factors. For example, an intra-patient variability of pazopanib trough levels and a decrease in plasma exposure over time have been demonstrated [3]. In addition, the co-administration of pazopanib with other xenobiotics may result in a modification of exposure. Indeed, previous studies indicate that pazopanib co-administered with esomeprazole causes C_max_ to decrease by 42% and AUC by 40% [8,9]. Furthermore, a significant correlation was found between P-gp inhibitors and dose reduction [10]. In addition, the administration protocol can also have an effect on pazopanib exposure; for example, when pazopanib is taken with a meal, a two-fold increase in both C_max_ and AUC is observed. Hence, taking 600 mg with food provides the same exposure as 800 mg in a fasted state [11,12].

All of these sources of variability make it very difficult to recommend the same dose of pazopanib for each patient. As a result, a more personalized approach to treatment is preferable in order to take these variabilities into account and thus offer the patient the best possible treatment. It is for this reason that a pharmacokinetically guided individualized dosing approach for pazopanib is not only an interesting way to enhance the efficacy of the treatment but also to prevent toxicity [13,14,15,16].

Based on this information, in clinical practice, a personalized approach to the dose selection is available for the treatment of each patient. Thanks to therapeutic drug monitoring (TDM), the C_trough_ of pazopanib in the blood is monitored and then compared to a reference value. After a comparison of the observed results with the reference value, the amount of pazopanib administered to the patient can be re-assessed and adjusted. Indeed, there is a high standard of documented proof for the correlation of C_trough_ and efficacy. A C_trough_ > 20.5 mg·L^−1^ is correlated with the progression-free survival for metastatic renal cell carcinoma (mRCC) and soft tissue sarcoma. Regarding toxicity, the standard of proof remains lower, but some studies demonstrate correlation or trends between C_trough_ and dose-limiting toxicity [7,17].

However, on a daily basis, in hospitals, it is not always easy to obtain the trough concentration of pazopanib from patients. Indeed, measuring this trough concentration must be carried out at a very specific moment, namely, just before the next administration of pazopanib. It is therefore essential not only to have perfect patient compliance (the pazopanib administration must be at the right time, i.e., after the C_trough_ measurement) but also it is essential that the patient be available at the time of blood sampling. Unfortunately, complying with these two conditions is sometimes difficult in clinical practice.

Another option to track patients’ exposure is the use of a limited sampling strategy (LSS); a specific number of samples collected at specific time points can provide a valid AUC in an individual patient. However, the use of AUC in TDM needs to be based on the validated schedule of blood sampling and not samples collected at any time. In addition, for TKi (and non-hospitalized patients in general), patients would have to come to the hospital several times a day, or because of the relatively long half-life of this class of drugs, they would have to stay at the hospital for several hours, which makes this procedure less convenient.

In this context, since the routine monitoring of trough plasma concentrations is difficult to implement, we aimed to combine a specific form of LSS with a modeling approach in order to determine the pazopanib exposure of a patient after a sample taken at any time. To achieve this goal, we decided to focus on the AUC based on the clearance (Cl). A popPK model was constructed in order to estimate individual apparent clearance (Cl/F) from a single point taken at a random time, then by using the dose (AUC = Dose/Cl/F), the AUC of each patient was calculated. We compared the AUC of each patient to their trough plasma concentration in order to define a target AUC corresponding to a trough concentration over 20.5 mg·L^−1^ [18].

## 2. Results

Pazopanib PK was best described as a monocompartmental model. The volume of distribution of pazopanib was V/F = 22.3 L (R.S.E.: 9.25%), a first-order absorption phase was modeled with an absorption rate, k_a_ = 0.976 h^−1^ (R.S.E.: 12.3%) and a first-order elimination phase was chosen with an apparent clearance, Cl/F = 0.458 L·h^−1^ (R.S.E.: 9.73%). ASAT was set up as a covariate and helped to decrease the inter-individual variability of V/F from 39.8% to 24.8% (Table 1).

Individual and population concentration versus predicted concentration was observed; the population prediction shows a slight overestimation of low drug concentrations and underestimation of high drug concentrations (Figure 1).

The popPK model was validated by observing the visual predictive check (VPC), with n = 1000 simulations (Figure 2), and the residual error and the NPDE were observed (Figure 3). A sinusoidal distribution of IWRES up to 8 h was observed, which is in line with the population prediction profile observed in Figure 1.

The distribution of parameters is also presented (Figure 4); a shrinkage value of 10.5%, 3.03%, and 11.5% was observed on ka, Vd/F, and Cl/F, respectively.

No correlation was observed between random effects. In addition, the distribution of random effects follows a normal law (Figure 5).

Pearson’s product-moment correlation test was performed between C_trough_ and the AUC of each patient, and a correlation factor of 91.9% was shown. This means the use of this AUC as a target, with this model, is similar to use a C_trough_ of 20 mg·L^−1^.

The individual apparent clearances estimated using the model were then used to calculate the AUC for each patient. In order to calculate a target AUC, we selected patients whose C_trough_ was greater than 20 mg·L^−1^. A target AUC of 750 mg·h·L^−1^, based on the target C_trough_, was chosen with a specificity of 90.6% and a sensitivity of 99.1% (Table 2).

A comparison was performed between the estimated AUC and Cl/F vs. real AUC and Cl/F with cross-validation leave-one-out method (Table 3). During this test, we observed that 48 times over 56, the AUC is under 750 mg·h·L^−1^ when the C_trough_ is under 20.5 ug·mL^−1^.

A Wilcoxon–Mann–Whitney statistical test was performed, and no significant difference was shown between Cl/Fs and AUCs obtained with the model and with Phoenix^®^ (*p*-value of 0.6305 for Cl/F comparison, *p*-value of 0.7394 for AUC comparison).

A bootstrap analysis was also performed in order to observe the accuracy of estimated pop PK parameters; 100 iterations were performed. All the popPK parameters of the model were between the 1st and 3rd quartiles.

Variation of AUC between cycle 2 and cycle 4 for patients from the clinical trial (NCT: 02331498) of pazopanib in combination with temozolomide, were observed. A statistically significant decrease in AUC between cycles 2 and 4 was demonstrated (Wilcoxon–Mann–Whitney test with a *p*-value of 0.0097) (Figure 6).

## 3. Discussion

The purpose of our paper is to determine a new method using AUC and show it as a possible new way to control pazopanib exposure. In this study, the correlation between AUC and Ctrough is suitable; as seen before, a correlation factor of 91.9% was shown between the AUC and Ctrough of each patient. This result gives the possibility to use the target Ctrough as a surrogate of evidence for the AUC’s usefulness.

Today, with therapeutic drug monitoring, it is possible to monitor precisely the evolution of xenobiotic compounds in a patient’s body. This approach makes it possible, in clinical practice, to personalize the medical treatments administered and therefore offers patients the best possible treatment. Monitoring plasma concentrations, or other parameters, in order to suggest dose adjustments, is a modern approach of medicine that gives patients access to high-quality health care, and therapeutic drug monitoring is an effective tool to optimize treatment by continuously ensuring its efficacy and preventing side effects [13,17,19].

TDM is an essential solution, particularly for compounds such as pazopanib, which, like many tyrosine kinase inhibitors, does indeed exhibit a large pharmacokinetic inter-individual and intra-individual variability. Indeed, an inter-occasional variability was observed for Cl/F and V/F, and a statistically significant decrease between cycle 2 and cycle 4 was shown for AUC.

These inter and intra-patient variabilities are one of the reasons why treatment failure is often observed. In oncology, in particular, it is indeed not uncommon to encounter drugs with a narrow therapeutic window, such as MKIs and pazopanib. For these compounds, it is important to administer the most suitable dose in order to optimize the therapeutic effects and, above all, to minimize the toxic effects. In the case of pazopanib, TDM makes it possible to monitor the C_trough_ and to compare it with a reference value in order to be able to adjust the amount of pazopanib to be administered subsequently.

There are several methods that can be used to make dosing adjustments of pazopanib. For example, because its metabolism is mainly hepatic, an adjustment of the administrated amount is made for patients with hepatic dysfunction. Hence, the dose of pazopanib can be reduced to 200 mg per day in patients with moderate hepatic impairment, and the drug is not recommended for patients with severe hepatic failure [20,21].

In order to adapt exposure, the dose can be split, and 400 mg of pazopanib can be administered twice a day, morning and evening [12,22]. Pazopanib can also be taken with food to increase exposure [23].

The trough concentration is the most convenient parameter to estimate exposure since it is necessary to obtain only one sample. Moreover, a correlation between C_trough_ and efficacy has been demonstrated for pazopanib both in GIST and mRCC, which consolidates the impact of the C_trough_ in the care of a patient. However, in clinical practice, the availability of a trough concentration is not so easy to implement, mostly due to the difficulty for the patients to take the drug at a defined time and to perform the blood sampling just before the next administration.

Hence, we aimed to find a way to predict pazopanib exposure from blood samples taken at any time by using a modeling approach. Developing an efficient popPK model for pazopanib is a challenge. In 2014, Imbs et al. developed a popPK model of pazopanib in order to study the variation of pazopanib PK administered in combination with bevacizumab and showed an inter-individual and inter-study pharmacokinetic variability that shows the need for further evaluation of therapeutic drug monitoring for pazopanib [24]. In 2017, Yu et al. developed a bi-compartmental (two-compartments) model to describe the complex absorption process, the non-linear dose–concentration relationship, and the high inter-patient and intra-patient variability [25].

Our team constructed a population pharmacokinetic model to estimate the individual clearance of the drug regardless of the time of sampling. A median Cl/F of the population of 0.46 L·h^−1^ was estimated. Once the clearance is determined and the administered dose is known, then it is possible and simple to estimate the patient’s AUC applying the well-known formula (AUC = Dose/(Cl/F)). Specificity and sensitivity tests make it possible to determine the target AUC of 750 mg·h·L^−1^ from the patient’s C_trough_. The comparison of the estimated AUC with the target AUC would make it possible to determine whether plasma exposure is adequate or whether it would be necessary to propose therapeutic dose adjustment. Moreover, as exposure decreases over time, TDM should be performed at different times to ensure long-term efficacy.

With this developed popPK model, we can have estimated the Cl/F and the AUC of a patient from a single point sample with an average deviation under 30%. Even if the estimation of the real AUC with this method is not perfect, it is important to remind that the purpose is not to have an exact estimation of the AUC. The goal of this approach is to calculated an AUC < 750 mg·h/L when the Ctrough < 20.5 ug/mL. After cross-validation with the leave-one-out approach, we observed that in more than 85% of cases (48 times over 56), the estimation of an AUC under 750 mh·h/L matches with a Ctrough under 20.5 ug/mL. In addition, it is important to notice that one part of the variability observed between the estimated AUC and Cl/F vs. real AUC and Cl/F is due to the method of calculation (linear up log down approach of Phoenix vs. Dose/Cl/F of the model).

However, deeper testing with a statistical approach is needed in order to validate this new approach and be able to use it in clinical practice.

The use of PK modeling is then an efficient approach to determine the level of pazopanib exposure for patients receiving oral targeted therapy and who come to their clinical center occasionally for a clinical visit

## 4. Materials and Methods

### 4.1. Data Set

The pharmacology laboratory of the Gustave Roussy Institute offers therapeutic drug monitoring (TDM) for many MKIs as the standard of care. A database of patients treated with pazopanib between 1 January 2012 and 27 December 2018 (n = 58) was created. The data set contains 55 adult patients and 3 children treated for soft tissue sarcoma and Ewing sarcoma with pazopanib at 200 to 800 mg·day^−1^. The trough pazopanib plasma concentration of each patient was measured at a mean of 24.6 h post-dose, and 126 samples were measured. Each patient was sampled between 1 and 6 times (mean = 2.2). In our population, 52 patients were administered pazopanib under fasted conditions, while 15 were administrated pazopanib with food. For 6 patients, the fasted state was unavailable. The following data were systematically retrieved: pazopanib dosing (mg), measured pazopanib concentration (µg·mL^−1^), sex, age, creatinine (µmol·L^−1^), albuminemia (g·L^−1^), aspartate transaminase (ASAT), and alanine transaminase (ALAT).

In addition to TDM of patients treated at Gustave Roussy, data from a previous clinical trial (NCT: 02331498) of pazopanib in combination with temozolomide were added to the database. This study was carried out on 15 adult patients treated for glioblastoma, for whom the full kinetic profile (over 24 h) of pazopanib was available. Pazopanib was administered at 200, 400, 600, and 800 mg/day. Temozolomide was administered at 1 to 200 mg/m^2^/day during 6 four-week cycles. Our laboratory analyzed plasma concentration of pazopanib in August 2018 for administrations at day 1 of cycles 1, 2, and 4, at 0, 0.5, 1, 2, 4, 6, 8, and 24 h post-dose. For 13 of the 15 patients, there were all 8 samples. For 1 patient, there were 7 samples, and for 1 other patient, there were 5 samples. A total of 280 samples were measured, and 36 kinetic profiles were observed.

Finally, the database included 406 samples: 36 kinetic profiles of 5 to 8 samples in 15 patients from the clinical trial and data from routine monitoring of 58 patients with 1 to 6 samples per patient. Missing covariate values for a patient were replaced by its patient median value; if no median value was available for this patient, the median population value was used (Table 4).

### 4.2. Sample Analyses

Blood samples were collected in lithium heparin tubes and were then centrifuged at 5000× *g* for 10 minutes. The plasma samples were stocked at −20 °C and were analyzed using high-performance liquid chromatography (HPLC) coupled with a UV detector (λ = 310 nm) with erlotinib as the internal standard. The lower limit of quantification is 1 mg·L^−1^ leading to a calibration range set between 1 and 100 mg·L^−1^. The method was validated according to the EMA guidelines with repeatability ranging from 0.89% to 1.74% and an accuracy ranging from 2.33% to 5.47% for our 4 QCs (1.5, 3, 15, and 90 mg·L^−1^) [26].

### 4.3. Pharmacokinetic Analysis and Model Building

Microsoft Excel^®^ 2012 software was first used to organize and merge the Gustave Roussy and Nice clinical study databases. The pharmacokinetics of pazopanib were analyzed using a non-compartmental method (NCA) with the Phoenix 64 WinNonlin^®^ software (Certara, NJ, USA), with linear up log down method, and the selection of half-lives were performed using the “best fit” option. Datxplore^®^ Lixoft^©^ version 2018R1 software (Lixoft, Anthony, France) was used to visualize the data by plotting the concentrations as a function of time and the logarithm of the concentrations as a function of time. Monolix^®^ version 2018R1 (Lixoft, Anthony, France) was used to build the population pharmacokinetic model.

The estimation of the population parameters was performed using the stochastic approximation expectation-maximization (SAEM) algorithm.

We were able to test several models, compare them and prioritize them according to their degree of likelihood calculated with a Monte Carlo size of 10,000, a degree of freedom fixed at 5, and without a linearization method. Models with 1, 2, and 3 compartments were tested, with first-order oral absorption, with or without lag time and linear elimination. The minimization of −2 × Log (likelihood), which was presented as the objective function value (OFV), was used to decide between models. A decrease in the OFV of 3.84 (*p*-value = 0.05) was considered significant.

The variability has been described using a model with a normal distribution and a lognormal distribution for the parameters. The residual variability is described using a combined model. Additionally, an inter-occasional variability was observed on Cl/F and V/F:
(1)$$\log(V/F)=\log(V_{\mathrm{pop}}/F)+\eta_{V_{i}}+\eta_{V_{\mathrm{occ}}}$$
(2)$$\log(\mathrm{Cl}/F)=\log({\mathrm{Cl}}_{\mathrm{pop}}/F)+\eta_{{\mathrm{Cl}}_{i}}+\eta_{{\mathrm{Cl}}_{\mathrm{occ}}}$$


With is the typical value of apparent volume for a patient, η_V_i__ and η_Cl_i__ the inter-individual variabilities (IIV) and η_V_occ__ and η_Cl_occ__ the inter-occasion variabilities (IOV).

Subsequently, covariates were added to the model by using the stepwise approach to refine the results obtained and reduce the observed variability. The effect of dose, age, sex, creatinine, albuminemia, ASAT, and ALAT was tested on all variables. Statistical significance of the covariates was evaluated on the basis of the Akaike information criterion (AIC):
(3)$$\mathrm{AIC}=\mathrm{OBJ}+2.n_{p}$$
where OBJ is the objective function of the model and n_p_ is the total number of parameters.

A decrease of at least 2 in the AIC was required for a covariate to be considered as significantly linked to the PK variable. Only ASAT showed a significant decrease in AIC with a significant *p*-value on V/F and was kept as a covariate in the final model as follows with a β_V/F_ = − 0.838 (29%):
(4)$$\log(V/F_{i})=\log(V_{\mathrm{pop}}/F)+\beta_{V/F}\cdot \log(\frac{{\mathrm{ASAT}}_{i}}{36.5})+\beta_{V_{i}}$$

With a median ASAT value of 36.5 UI·L^−1^.

The statistical model describes how V_i_/F is distributed around these predicted values:
(5)$$\log(V_{i}/F)∼N(V_{i}/F),{\omega_{V/F}}^{2})$$


Evaluation of the final model that had been selected was performed using the visual predictive check (VPC) method. VPCs were obtained on Monolix^®^ with a total of 1000 replicates were simulated, using the final model to simulate expected concentrations, and the 90% prediction intervals were generated. The observed data were overlaid on the prediction intervals and compared visually, and the binning criteria were least-square. No stratification was performed.

A bootstrap analysis with 100 iterations was performed. The observations show that all the popPK parameters estimated by the model are between the 1st and 3rd quartile.

Ten patients from the clinical trial with full PK profiles were used to validate the model and to compare the estimated AUC and Cl/F vs. real AUC and Cl/F. Cross-validation with the leave-one-out approach was performed. One of the ten patients at steady state was removed from the data set, and then the model was built on this new data set with n − 1 patients. The left-out patient was then used to test the model: his Cl/F was calculated for each of his 8 samples individually, the AUC was calculated from these Cl/F and compared to the real AUC and Cl/F from Phoenix. This approach was repeated 10 times for all patients at a steady state.

The study was conducted in accordance with applicable laws and regulations, and the declaration of Helsinki, with the approval of the Institutional Review Boards (IRB).

GraphPad Prism 6^®^ software (GraphPad, California, USA) was used to perform specificity and sensitivity tests in order to determine the target AUC threshold, representative of a patient with the C_trough_ of 20 mg·L^−1^.

R^®^ software (R Core Team, Vienna, Austria) was used for statistical comparison of AUC between cycle 2 and cycle 4 of patients from the clinical trial. A Wilcoxon–Mann–Whitney test with paired samples was carried out, and a significant *p*-value of 0.0097 was observed.

## 5. Conclusions

TDM and personalized medicine are modern approaches to healthcare that aims to increase the effectiveness of treatments administered to the patient while reducing risks and taking account of intra- and inter-patient variability. In order to make this approach possible, it is essential to select the most suitable parameter to monitor the patient. It is also necessary to determine reference values of these parameters, which can be compared with the assay results, and thus to propose a dose adjustment for the patient. However, it is not always possible to obtain the desired parameters due to the many constraints linked to patient compliance or to blood sampling times.

In this project, we observed that the use of pharmacokinetic modeling with a population approach could solve this problem by estimating the AUC of a patient. We were thus able to determine whether the level of pazopanib exposure was correct or not, regardless of the time of sampling.

The combination of TDM and pharmacokinetic modeling would seem to be a suitable choice to improve the efficiency of the treatments administered to patients in the future.

## Data Availability

Not available.

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
