# Peer review of "Population Pharmacokinetic Analysis of Pazopanib in Patients and Determination of Target AUC"

_pharmaceuticals, 2021, doi:10.3390/ph14090927_

Round 1
Reviewer 1 Report
General comments
The manuscript Pharmaceuticals-1271891 describes the development of a population PK model for pazopanib in cancer patients. Also, it attempts to create a strategy for determination of drug AUC basing on plasma samples collected at any (convenient) time points, intended for use in TDM instead of Ctrough measurement. The paper is generally well-organized and written in good English. However, in my opinion, the paper has serious flaws – it has a poor scientific value, relies on wrong assumptions and simplifications, and provides a weak novelty:
- According to the Authors, the rationale behind the development of the AUC-based TDM of pazopanib is that a timely collection of blood for Ctrough determination is difficult in clinical practice. I find this claim unjustified and strained. In a Ctrough approach, the hospital staff, at the same “meeting” with a patient, collect one sample of blood and provides/controls administration of the drug administration. In an AUC approach, the staff need to collect more than one sample (usually 2-4) and, if drug administration is scheduled at different time, also need to visit the patient once again. For that reason, measuring Ctrough has been for decades a first-choice option in TDM.
- Indeed, for some drugs, the use of AUC in TDM is preferred over Ctrough, when it has been proved that the AUC better correlates with clinical outcomes. According to the current clinical data, mentioned also in the manuscript, the efficacy and toxicity of pazopanib relate well to the Ctrough of pazopanib. In opposition, the data on AUC vs. clinical outcome relationship are scarce. Therefore, at present, the use of pazopanib AUC in TDM is unjustified.
- Leaving the relevance of AUC in TDM of pazopanib aside, the determination of the drug AUC from blood samples taken at any time is an oversimplification. Using one of the methods of a limited sampling strategy (LSS), it should be proved that the specific number of samples collected at specific time points provides a valid AUC in an individual patient. The use of AUC in TDM should be based on the validated schedule of blood sampling, not samples collected at any times, which makes this procedure less convenient than the Ctrough measurement.
- The manuscript does not add novel elements compared with the 4 already published population PK models for pazopanib. The presented model is very similar to the one reported by Imbs et al. (Cancer Chemother Pharmacol. 2014;73:1189–1196).
- Moreover, neither in the Introduction nor Discussion, the Authors refer to/mention the previous population PK models developed for pazopanib in humans: Imbs et al., Cancer Chemother Pharmacol. 2014;73:1189–1196; Yu et al., Clin Pharmacokinet (2017) 56:293–303; Sternberg et al., Clin Cancer Res. 2018 ,24(13): 3005–3013; Baneyx et al., PAGE 26 (2017) Abstr 7105 [www.page-meeting.org/?abstract=7105]. Therefore, the essential state-of-the-art related to the paper topic is not presented.
- The quality of the developed population PK model is average:
- The study has a retrospective character and the model was built basing on the two completely different datasets that were pooled together: 58 patients, both children and adults, in whom only the Ctrought was measured as part of TDM, and 15 clinical trial adult patients from whom 35 full concentration profiles were obtained. Another thing is a heterogeneity of the two groups, for example with respect to age (children only in the first group). All these make the two datasets were very imbalanced (for example: for children, only the concentrations at 24 h after dosing were available), which reduces the population PK model robustness
- some patients received the drug in a fed state. Despite it is known that a food state affects the pazopanib PK, the food state was not a covariate in the final model
- moreover, neither age nor body weight was a covariate in the final model. This is very surprising having the patients between 3 and 87 years of age
- Figures 1 and 3 show that the model overestimates low drug concentrations and overestimates high drug concentrations. In particular, it is seen from a sinusoidal distribution of the IWRES up to 8 h
- building of the population PK model in Monolix is weakly described. More details of the whole procedure should be given. The same concerns the noncompartmental analysis in WinNonlin.
- Introduction, Discussion, and Conclusions contain mostly general knowledge/statements, instead of being focused on a specific problem. Moreover, the Discussion seems to be suddenly cut at the end.
- The figures presentation needs improvement. The units are missing, and the figures and fonts are so small that are hardly readable
Specific comments
Line 38: it is worth adding that the pazopanib tablet was an IR tablet
Line 39: phase I clinical trial (Ref. #2) is not a good source for the recommended dosing of the drug. Instead, a final phase III trial or EMA/FDA assessment report should be cited
Line 40: more up-to-date PK papers are expected as a reference for the drug Cmax, instead of the paper from an early stage of drug development. Moreover, the type of dosing/dose that corresponds to the Cmax should be given
Line 42 and elsewhere: I am not sure if this format of units with dots at the bottom (µg.h.mL -1) is correct for the journal
Lines 51-52: at the end of the sentence, it should be add that the described steady-state occurs after a once-daily dosing (if this is true)
Line 54: the described Ctrough-PFS relation was observed in renal cancer patients, but not sarcoma patients
Line 56: its seems that Ref. #5 did not study the PK of pazopanib, therefore is not a good source here
Line 64: Ref. #7 is not a PK paper, so does not match here
Line 78 and elsewhere: Does the “residual concentration” mean a Ctrough. If so, it should be replaced with “Ctrough”
Line 85 and References: Ref. #16 is the same as Ref. #6 (doubled)
Lines 131-132: instead of stating “also the distribution of random effects was observed”, I suggest giving the results/interpretation of this observation
Lines 144-147: I do not understand the connection between the table content and its footnote
Lines 201-220: The number of children, dose of pazopanib, and diseases being an indication of pazopanib treatment are expected to be shown for the two groups of patients from whom the dataset was obtained for building the population PK model
Line 234: “low limit of quantification” should be replaced with : “lower limit of quantification”
Line 237: I have not met the term “fidelity” with respect to the analytical method validation. The presented data are the accuracy expressed as a relative error of determination
Table 1: I suggest to explain in the table footnote that the IIV is expressed as omega squared (as I guess). Moreover, the values of likelihood, AIC, and BIC mean very little if not compared to the values for the other model, so I suggest removing them from the table
Table 3: the unit of age (years) is missing
Figure 2: Selection of bins for VPC needs improvement. The number of points enable to use more than 4 bins.
Thank you
Author Response
Please see the attachment.
Thank you very much for your review, please do not hesitate to share your thoughts and questions.

Reviewer 2 Report
Paper seems to be well written and clear. The focus of this research is important and interesting. For these reasins, only minor revisions (see file enclosed) are necessary in my opinion.

Author Response
Please see the attachment.
Thank you very much for your review, please do not hesitate to share your questions and thoughts.

Reviewer 3 Report
This work quite topical and the overall story while very short is well written and coherent; with some insights into Pazopanib pharmacology profile and the application in the clinic.
The manuscript has some issues -
The presentation and content of some figures detracts from the main manuscript.
Figure 2 - can this be believed with no data post-25 hours? This prediction is not clearly explained.
Figure 3, 5 and 6 are poorly formatted and need to be revised.
Figure 6 - box and whisker graph should be improved to a publication quality image.
Line 13 - Pazopanib is not 'selective', this needs to be removed.
I would like to see a little bit more drug discovery discussed in either the introduction of the discussion and relating that to this study. I think this would be good for the reader and put this work into context.
Here are some references that could be considered for that section -
https://pubmed.ncbi.nlm.nih.gov/24224933/
https://pubmed.ncbi.nlm.nih.gov/30221034/
https://pubmed.ncbi.nlm.nih.gov/24727486/
https://pubmed.ncbi.nlm.nih.gov/31698822/
https://doi.org/10.1016/j.ejps.2019.01.010
https://doi.org/10.1124/dmd.108.024075
In addition, this landmark review publication should be mentioned in the introduction paragraph - https://www.nature.com/articles/s41573-021-00252-y
The authors should thoroughly re-read this manuscript and provide a clean version for final review.
Once these modifications have been done, this work should move forward.
Author Response
First of all we would like to thank the reviewer for his/her help.
We understand the reviewer's request. However, the main topic of our study is the use of PK parameters in order to improve the efficiency of pazopanib therapeutic administration. From our point of view, drug discovery, structure-activity relation and molecular modelization of TKIs would confuse the reader and give a misinformation about our focus, which is the estimation of CL in order to obtain the AUC of pazopanib.
Fixed issues :
Figure 2 was modified
Figure 3, 5 and 6 were reworked
Line 13 The word “selective” was removed
Round 2
Reviewer 1 Report
The paper has been improved to a very small extent. In general, I maintain my major concerns about it.
- I can understand the Authors’ explanations about the logistics problems in the determinations of the pazopanib Ctrough in an outpatient care. Obviously, it is much more convenient to measure a random concentration during the patient’s visit to the clinic facility.
- There are no clinical reports about the correlation between the efficacy/toxicity of pazopanib and its AUC, whereas they are such with respect to the Ctrough. The Authors argued that TDM of carboplatin and busulfan is based on AUC, but these are the drugs for which the efficacy-AUC relationship has been proved, in particular for busulfan. Advantageous for the paper is a good correlation found between the target AUC and target Ctrough, which might be eventually treated as a surrogate of evidence for the AUC usefulness. However, this would need discussion in the paper.
- I maintain my comments about a poor description and quality of the developed popPK model. It has not been improved in any way. I encourage the Authors to read some popPK papers to find how the building of the model is described. I give some examples in the point 5 below.
- Figure 2 still needs improvement as the number of bins in the VPC graph is very small (4 only) in relation to the number of available points. Therefore, the graph looks strange. More bins are expected.
- Definitely, the major flaw of the paper is the estimation of the pazopanib clearance in an individual patient (and then AUC = D/(CL/F)). This is only mentioned in few words in the manuscript, and readers will have no idea how it was performed and what results were obtained. From the Authors’ responses to the review, I am now guessing (but I am not sure) that their intention was to estimate the clearance (and then AUC) from a single measurement of the drug concentration at a random time point (one blood sample only). Taking into account the huge IIV of clearance of pazopanib in the studied population (omega squared as big as 0.714, which means CV% about 100%!), the use of one sample for that purpose seems risky, but maybe possible. I can only guess that this was done using Bayesian methods. Irrespective of the method used, it must be described in details in the Methods section, together with the mode of validation of the estimator. Moreover, in the Results section, the results of AUC (obligatory) and CL (optional) that were estimated from limited sampling (e.g., one blood sample) must be compared with the real AUC or real CL. In other words, the Authors need to prove that their estimation method is reliable. I think that a helpful paper in this respect would be:
Gotta et al. Therapeutic drug monitoring of imatinib: Bayesian and alternative methods to predict trough levels. Clin Pharmacokinet 2012; 51 (3): 187-201 (in particular Figures 6-9 and Table III).
The other examples that show the development/description of popPK models and Bayesian estimators are:
Benkali et al. Population pharmacokinetics and Bayesian estimation of tacrolimus exposure in renal transplant recipients on a new once-daily formulation. Clin Pharmacokinet 2010; 49(10):683-92.
Yu et al. Population pharmacokinetics and Bayesian estimation of mycophenolic acid concentrations in Chinese adult renal transplant recipients. Acta Pharmacologica Sinica 2017; 38, 15661579
Kassir et al. Population pharmacokinetics and Bayesian estimation of tacrolimus exposure in paediatric liver transplant recipients. Br J Clin Pharmacol. 2014 ;77 (6):1051-63.
Alsultan et al. Limited sampling strategy and target attainment analysis for levofloxacin in patients with tuberculosis. Antimicrob Agents Chemother 2015; 59: 3800-3807.
Ralph et al. Maximum a posteriori Bayesian estimation of epirubicin clearance by limited sampling. Br J Clin Pharmacol. 2004; 57(6): 764-772. - As shown in the mentioned paper by Gotta et al. (Clin Pharmacokinet 2012; 51 (3): 187-201), the estimation of the Ctrough from a single measurement at random time is possible, therefore the Authors may consider this approach instead of AUC estimation
- Because of incomplete bioavailability of oral pazopanib, the apparent CL (CL/F) should be used instead of just CL (also in the formula for AUC). Now the term “AUC/F” is used in the paper, which is inappropriate. “AUC” only should be presented, as it directly comes from the concentrations. CL and Vd are the parameters that incorporate the bioavailability (F), and then they are called “apparent”. Moreover, it is expected to indicate that the measured AUC relates to a steady state (AUCtau).
- The Discussion is superficial. The Authors argue that “after discussion with editors, they specifically asked us to add some general knowledge and statements”. However, this does not prevent giving the Discussion more deepness, especially in terms of the methodology, reliability/validity, and clinical relevance of pazopanib AUC or Ctrough estimation from limited sampling.
- In the Introduction, it is worth to mention the usefulness of limited sampling-based TDM for TKIs therapies.
Minor concerns:
Lines 46-47: The dose corresponding to the given AUC should be specified
Line 117: IIV means interindividual, not intraindividual, variability
Line 121: in “underestimation high” the “of” is missing
Line 126: “IWERES” should be replaced with “IWRES”
Line 212: “-ou” is a typing error
Line 276: “the NCA analysis was done using phoenix software” is unnecessary as it repeats the text from the line above.
Lines 349 and 413: more details for these two references are required: the website address and date of the last access.
Figure 2: Selection of bins for VPC needs improvement. The number of points enable to use more than 4 bins.
Figures 5 and 6 still need a bigger front in the labels to be readable.
Thank you
Author Response
We would like to thank the reviewer for his/her help.
Please see the attachment for the answers.

Reviewer 3 Report
The authors should include this landmark review publication in the introduction paragraph - https://www.nature.com/articles/s41573-021-00252-y. There is no confusion, a wider context to this work is important. Once this is done the paper is ready to go.
Author Response
We would like to thank the reviewer for his/her support. Indeed the following source is a good update.
The followin paragraph was added at the beggining of the introduction:
"Currently there are 88 small molecule kinase inhibitors (SMKIs) approved by the FDA and by other regulatory agencies. Their main application area is oncology, however one-third of the SMKIs in clinical development address different kinds of disorders such as rheumatoid arthritis. Latest clinical trial of SMKIs shows that the approximately 45 targets of approved kinase inhibitors represent only about 30% of the human kinome, which indicates that there are still substantial unexplored opportunities for this drug class[1]."
[1]. Attwood, M.M.; Fabbro, D.; Sokolov, A.V.; Knapp, S.; Schiöth, H.B. Trends in Kinase Drug Discovery: Targets, Indications and Inhibitor Design. Nat. Rev. Drug Discov. 2021, doi:10.1038/s41573-021-00252-y.
Best regards.
Round 3
Reviewer 1 Report
At now, I am getting the overview of what was done in the paper. It comes from the revised manuscript that a Bayesian approach was used to estimate an individual CL/F, and then AUC, which was unclear in the previous version. The paper has still some major flaws, most of them concerning the validation of the AUC estimation (prove of reliability):
- First of all, the validation of the AUC estimation from one sample is negligible. Table 3 presents the comparison of only 10 pairs of observed and estimated AUC, which is not convincing. Validation must be much more thorough and sophisticated. As the intention is to estimate the CL/F from one concentration measured at any (not predefined) time at steady state, the Authors should test dense time points from the whole dosing interval, in particular those that differ from the sampling times that were used for the popPK model development (e.g., every 0.1 h in at least 10 patients would make an appreciable dataset). The missing “observed” concentrations at these dense time points can be simulated using the final popPK model (obviously, including the residual error model).
- The performance of the AUC estimation should be evaluated using PE%, MPE, and RMSE to show the precision and accuracy of the estimator. The number of individual estimates outside the +/- 20% PE should be also indicated.
- It should be proved how the estimator works for the same subject, when CL/F is estimated from different sampling times - no significant differences in the CL/F and AUC are expected.
- A particular weakness of the current AUC estimator validation is that the prediction is shown only for the subjects that were used for the popPK model development, which does not reflect a real clinical situation. The reliability of the estimator should be proved with one dataset as a test group (learning group for model development) and the other as a validation group (for AUC estimation validation). The Authors may divide 36 kinetic profiles from the clinical trial patients in such a way. An alternative solution would be simulating virtual patients resembling the population studied in terms of ALAT level distribution and dividing their kinetic profiles into two groups, or doing a cross validation, for example a leave-one-out cross-validation.
- In line 175, the Authors state that “time point was added to a new dataset as Ctrough”. I do not understand why the concentration measured at any time was treated as Ctrough. This is a wrong procedure. The concentration should be attributed to a specific time at which it was measured. This wrong procedure would burn all the AUC estimation done so far.
- The methodology of the validation of the AUC estimation should be described in details in the Methods section.
- Once the AUC estimation validation is expanded, the results of it should constitute an important part of Discussion.
- The description of the methodology of the popPK model development and individual CL/F estimation has been expanded but some details are still missing:
-
- The algorithm used to estimate the model parameters (in Monolix, it is SAEM only, but users of the other programs for popPK modeling may not know this)
- Was a log-likelihood (together with AIC) computed with or without a linearization method?
- The Authors state that AIC was used for decision-making about covariates inclusion. What criterion was used for deciding between a one-, two, and three-compartment model and Tlag inclusion?
Minor concerns:
Lines 101-108: Actually, what the Authors are doing is also a LSS, but in a specific form, with single sampling possible at any time. Please revise this fragment to avoid confusion. Also, the language style needs refinement (sentence starts with “Plus …”).
Line 113: please replace “each patients CL” with “individual CL/F”.
Line 114: “CL/F” should be used instead of “CL”.
Line 177: please replace “After running the model” with “After running the estimation” or “After running the estimation of model parameters”.
Line 181: “CL/F” should be used instead of “CL”.
Lines 260-266: This fragment much better matches the beginning of the Discussion, not its end.
Line 340: “akaike” should be written with an uppercase letter as it comes from the surname of prof. Hirotugu Akaike.
Table 3: a more readable numbering of “Subject ID” should be used, e.g., integers.
Figure 5: “Density” label in the Y-axis is hardly seen (small font).
Thank you
Author Response
We would like to thank the reviwer for his/her help and support.
Please see the attachment for the answers.
Best regards
